# Thermal Stress Simulation and Structure Failure Analyses of Nitrogen–Oxygen Sensors under a Gradual Temperature Field

**Jiangtao Feng [1], Jiaqi Geng [1], Hangyu She [2], Tao Zhang [2], Bo Chi [1] and Jian Pu [1,\*]**

[1] School of Materials Science and Engineering, Huazhong University of Science & Technology, Wuhan 430074, China; peterphon@lambdasensor.cn (J.F.); jiaqigeng001@163.com (J.G.); chibo@hust.edu.cn (B.C.)

[2] School of Naval Architecture & Ocean Engineering, Huazhong University of Science & Technology, Wuhan 430074, China; m202172173@hust.edu.cn (H.S.); zhangt7666@hust.edu.cn (T.Z.)

\* Correspondence: pujian@hust.edu.cn; Tel.: +86-27-8755-8142

**Abstract:** Nitrogen–oxygen sensors are pivotal for $NO_X$ emission detection, and they have been designed as key components in vehicles' exhaust systems. However, severe thermal stress concentrations during thermal cycling in the sensors create knotty structural damage issues, which are inevitable during the frequent start–stop events of the vehicles. Herein, to illustrate the effect of thermal concentration on a sensor's structure, we simulated the temperature and stress field of a sensor through finite element analysis. The failure modes of the sensor based on the multilayer structure model were analyzed. Our simulation indicated that the thermal deformation and stress of the sensors increased significantly when the heating temperature in the sensors increased from 200 to 800 °C. High stress regions were located at the joint between the layers and the right angle of the air chamber. These results are consistent with the sensor failure locations that were observed by SEM, and the sensor's failures mainly manifested in the form of cracks and delamination. The results suggest that both the multilayer interfaces and the shape of the air chamber could be optimized to reduce the thermal stress concentrations of the sensors. It is beneficial to improve the reliability of the sensor under thermal cycling operation.

**Keywords:** nitrogen–oxygen sensor; finite element analysis; simulation; temperature distribution; thermal stress concentration

## 1. Introduction

In recent years, the high level of nitrogen oxides ($NO_x$) in the atmosphere has raised serious concerns, because $NO_x$ can cause severe injuries and trigger respiratory diseases such as asthma and cardiovascular diseases after inhalation [1–3]. To reduce $NO_X$ emission, automotive industries have applied different strategies over the past two decades, and significant developments have been devoted to refining vehicle emission systems. As key modules, nitrogen sensors monitor the concentration of certain gases, such as NO and $NO_2$, emitted by fuel combustions [4–6]. When the $NO_x$ emission concentration surpasses a critical value, the nitrogen–oxygen sensor will send spontaneous feedback to the electronic fuel injection system to adjust the fuel–air ratio; therefore, $NO_x$ emissions can be reduced [7,8].

Among all the nitrogen–oxygen sensors, solid electrolyte nitrogen–oxygen sensors are the most popular because of their excellent stability and responsive performance at high temperatures. The structure of these sensors usually includes a zirconia based $O^{2-}$ conductor electrolyte with a sensitive electrode and a reference electrode arranged on its two sides [9]. When both $NO_x$ (NO or $NO_2$) and $O_2$ exist in the test atmosphere, the following electrochemical reactions will take place:

In a NO and $O_2$ atmosphere:

Sensitive electrode:

$$NO + O^{2-} = NO_2 + 2e^- \tag{1}$$

Reference electrode:

$$\frac{1}{2} O_2 + 2e^- = O^{2-} \tag{2}$$

In a $NO_2$ and $O_2$ atmosphere:
Sensitive electrode:

$$NO_2 + 2e^- = NO + O^{2-} \tag{3}$$

Reference electrode:

$$O^{2-} = \frac{1}{2} O_2 + 2e^- \tag{4}$$

The $O^{2-}$ involved in these reactions will migrate through the electrolytes and produce a responsive potential that can convert the $NO_x$ concentration signal to an electrical signal [10,11]. Reactions (2) and (4) indicate that the nitrogen–oxygen sensor exhibits opposite response potential signals when detecting NO and $NO_2$, because $O^{2-}$ moves in the opposite direction: the sensor produces a positive potential signal when detecting $NO_2$ and a negative potential signal when detecting NO. Thus, it is feasible for the sensor to differentiate NO and $NO_2$ concentrations.

In order to obtain a more responsive performance, it is important to improve the $O^{2-}$ conductivity of the $ZrO_2$-based electrolyte. As a response, a small amount of $Y_2O_3$ is proposed to be doped into $ZrO_2$ to form a cubic structure, called yittria-stabilized zirconia (YSZ) [12,13], as well to replace $Zr^{4+}$ with lower valence $Y^{3+}$, creating abundant oxygen vacancies that effectively improves the $O^{2-}$ conductivity of YSZ. On account of the advantages, the YSZ-based electrochemical cell has been broadly applied in solid oxide fuel cells (SOFCs) [14–17]. Similar to SOFCs in multilayer structures and materials systems, planar-type $NO_x$ sensors are fabricated by tape casting, screen printing, and high-temperature sintering techniques sequentially, which consist of multifarious ceramic layers and metallic electrodes [18]. $NO_x$ sensors usually operate at high temperatures, because YSZ electrolytes can only meet the $O^{2-}$ conductivity requirements at temperatures higher than 600 °C [19]. Integrated planar $NO_x$ sensors have been designed consisting of the heating and sensing regions that can achieve rapid signal response. However, such designs also increase the fracture probability of sensors at working condition, especially on the thermal cycling operation [20].

Usually, a gradual temperature field is inevitable for the heat transfer process from the engine to the sensors, which is generated from combustion reactions. Accordingly, uneven thermal stress distribution is generated on the multilayer ceramics inside of the sensor during the heating process. Not only could this thermal stress crack the ceramics, it can also cause other structural damage and deactivate the sensors. Currently, most studies focus on novel materials and the structural design of the sensors [21–23], and few reports exist on the mechanical properties and structural strength, especially when the sensors are experiencing thermal cycles [24]. Therefore, it is necessary to investigate the thermal matching properties of each layer in $NO_x$ sensors at high temperatures. However, it is hard to characterize these properties or stress states in such tiny devices. Finite element analysis (FEA) provides an effective method to simulate real physical systems using mathematical approximations, and it can be applied to various complex shapes with high calculation accuracy. A.H. Elsheikh used FEA to verify the thermal effects on the deflection and stresses in thin-walled workpieces [25]. M. Ahmadein adopted FEA as a standard to evaluate their simulation results, which were performed to obtain the relationship between diversified thermo-physical properties and the heat transfer process [26]. Both of them achieved good agreement between the experimental and numerical results. In this paper, the thermal stress FEA program was used by introducing the structural parameters of a planar $NO_x$ sensor, and the effect of heating temperature on the thermal stress distribution was revealed, which provides solid theoretical support and sheds light onto better structural designs

and functional optimizations for future NO$_x$ sensors. Based on the numerical simulation results, it can predict the type and location of the defect of the sensor under thermal cycling operation.

## 2. Experimental

### 2.1. Structure of the NO$_x$ Sensor

In this study, we selected a typical planar NO$_x$ sensor that consisted of a sensing and heating region for further temperature and stress analyses. Figure 1 shows the multilayer structure of the tested NO$_x$ sensor. The eight-wire structure was composed of a main pump, auxiliary pump, measuring pump, and heating circuit. The rectangular ceramic-based NO$_x$ sensor, with a 45 mm length and a 4 mm width, was fabricated with 8 layers of components, including the sensitive electrode—1; ZrO$_2$ functional layer—1; sensitive electrode—2; ZrO$_2$ air flow layer; sensitive electrode—3; ZrO$_2$ functional layer—2; Al$_2$O$_3$ insulating layer—1; heating circuit layer; Al$_2$O$_3$ insulating layer—2.

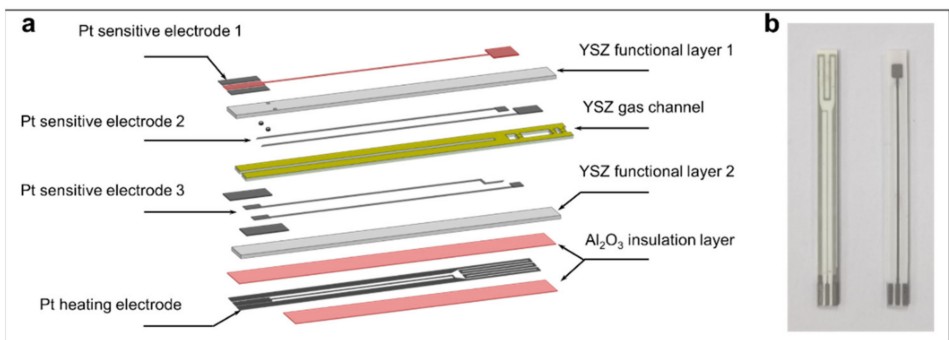

**Figure 1.** The planar nitrogen–oxygen sensor: (**a**) schematic structure via an exploded view of the sensor; (**b**) outside view of the two sides of the sensor.

### 2.2. Modeling and Mesh Generation

The tested model was built with 5 layers: (1) Al$_2$O$_3$ insulating layer, length × width × thickness = 45 mm × 4 mm × 60 µm; (2) the Pt heating circuit, length × width × thickness = 40 mm × 3 mm × 15 µm, which located in the middle of the Al$_2$O$_3$ insulating layer; (3) ZrO$_2$ functional layer—1, length × width × thickness = 45 mm × 4 mm × 0.4 mm; (4) ZrO$_2$ air flow layer, length × width × thickness = 45 mm × 4 mm × 0.4 mm and chamber structure of 0.4 mm × 2 mm; (5) ZrO$_2$ functional layer—2, length × width × thickness = 45 mm × 4 mm × 0.4 mm. It should be noted that the sensitive electrode layer with a thickness of 5 µm in the NO$_x$ sensor was not taken into account in the simulation to simplify the calculations. Additionally, because the NO$_x$ sensor has a highly symmetrical structure, the model was constructed to only one-quarter of the actual volume to shorten the computational time. The simplified model and related mesh generation of the NO$_x$ sensor are presented in Figure 2. The mesh size was 0.02 mm. With a total number of 120,000 grids and 510,000 nodes, different layers were labeled in different colors: ZrO$_2$—blue; Al$_2$O$_3$—purple; Pt heating circuit—red.

The finite element analysis was conducted using ANSYS R16.0 software. The grid model was first built with SOLID 70 units to generate a general approximation for the temperature distribution. It was assumed that the sensor was in a steady-state temperature field, and the Pt electrode was the internal heat source. The following equations were obtained:

$$\frac{\partial^2 T}{\partial x^2} + \frac{\partial^2 T}{\partial y^2} + \frac{\partial^2 T}{\partial z^2} + \frac{Q}{K} = 0 \tag{5}$$

where T is temperature; x, y, and z are the coordinates; K is the thermal conductivity; Q is the amount of heat generated by the internal heat source per unit volume. For the

multilayered composite structure, the thermal conductivity in the plane (directions: $x$ and $y$) and $z$-axis direction can be solved as:

$$K_{x,y} = \frac{k_1 d_1 + k_2 d_2 + k_3 d_3}{d_1 + d_2 + d_3} \tag{6}$$

$$K_z = \frac{d_1 + d_2 + d_3}{d_1 / k_1 + d_2 / k_2 + d_3 / k_3} \tag{7}$$

where $k_1$, $k_2$, and $k_3$ are the thermal conductivities of Pt, $Al_2O_3$, and YSZ; $d_1$, $d_2$, and $d_3$ are the thicknesses of the corresponding layers. Then, the adopted unit was transformed into SOLID 185, with which the thermal stress distribution could be analyzed based on the temperature distribution results. Assuming that no bending strain was in the multilayer structure sensor, the thermal stress of each layer can be expressed as:

$$\sigma_i = E_i (\varepsilon_i - \alpha_i \Delta T)/(1 - v_i) \tag{8}$$

where $\sigma_i$ and $\varepsilon_i$ represent the thermal stress and strain of layer i; $E_i$ is Young's modulus; $v_i$ is the Poisson ratio of the corresponding material; $\alpha_i$ is the expansion coefficient. After meshing the generations of the model, all layers in contact with air were set as convective heat transfer surfaces.

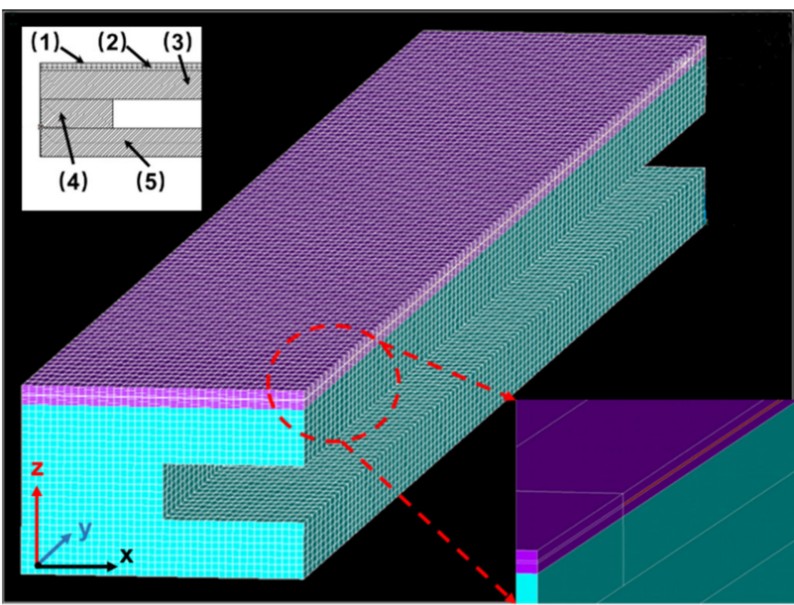

**Figure 2.** The quarter simplified model and related mesh generation of the nitrogen–oxygen sensor.

*2.3. Boundary Conditions and Parameters*

To simply the initial computation, the $NO_x$ sensor model was assumed to be in an adiabatic state; therefore, the heat dissipation of Pt heating circuit was not taken into consideration. Because of the highly symmetrical structure of the $NO_x$ sensor, the model was only one-quarter of the actual structure, symmetrical boundary conditions were applied on both sides of the model to provide better approximation. It was assumed that the ambient temperature and Pt heating circuit temperature were constant; then, the natural boundary conditions between the environment and the sensor can be expressed as:

$$-K (\text{gradT} \cdot n) = Q - h (T - T_0) - \lambda\mu (T^4 - T_0^4) \tag{9}$$

where n is the unit normal vector of the heat transfer surface; h is the convective heat transfer coefficient, which was set at 25 W/(m$^2$ K); $T_0$ is the ambient temperature, and

it was set at 20 °C; $\lambda$ is the thermal emissivity rate, and it was set at 0.4; $\mu$ is the Stefan–Boltzmann constant, $\mu = 5.67 \times 10^{-11}$ mW/(mm$^2$ K).

When determining the stress field distribution, the equations include thermal stress ($\sigma$) and thermal strain ($\varepsilon$). It was assumed that there was no bending strain in the sensor; thus, the resultant stress of the multilayer structure was 0 and the boundary condition can be written as:

$$\sum_{i=1}^{n} Ei \, (\varepsilon i \, - \, \alpha i \, \Delta T) \, / \, (1 \, - \, vi) = 0 \tag{10}$$

With a broad temperature range for the thermal expansion coefficient that varied from the ambient temperature to 800 °C, the physical properties of the ZrO$_2$, Al$_2$O$_3$, and Pt heating circuit used in the numerical simulations are illustrated in Table 1.

**Table 1.** Physical parameter settings of YSZ, Al$_2$O$_3$, and Pt heating circuit for numerical simulation.

| Parameters | YSZ | Al$_2$O$_3$ | Metallic Pt |
|---|---|---|---|
| Elastic modulus (GPa) | 210 | 350 | 169 |
| Bending strength (MPa) | 1100 | 2500 | 922 |
| Density (g/cm$^3$) | 6.00 | 3.85 | 21.45 |
| Coefficient of thermal expansion (K$^{-1}$) | $10 \times 10^{-6}$ | $8.2 \times 10^{-6}$ | $9 \times 10^{-6}$ |
| Thermal conductivity (W/(m·K)) | 22.0 | 25.0 | 71.4 |
| Poisson ratio | 0.30 | 0.23 | 0.39 |

### 2.4. Micromorphology Observation

The samples for the cross-sectional SEM characterization were embedded in Buhler epoxide and polished by a Buhler automatic polisher. The samples were observed by SEM (Quanta 200, FEI, Eindhoven, The Netherlands) with an energy dissipation spectrum (EDS) attachment. The X-ray micro CT (Zeiss XRM Xradia 520 Versa, Oberkochen, Germany), with a point resolution of 0.9 μm, was also used for defect observations in the samples.

## 3. Results and Discussions

### 3.1. Temperature and Thermal Stress Distribution Based on Numerical Simulation

The related temperature distribution under different heating temperatures was calculated to simulate the thermal stress distribution in the NO$_x$ sensor operation. Due to the frequent engine starts and stops, the NO$_x$ sensors usually work in wide temperature ranges. To better approximate such operational environments, multiple temperature points were selected to simulate the thermal fields. Figure 3a–d represent the temperature distributions of the Pt heating circuit at four different temperatures, ranging from 200 to 800 °C. It could be found that the temperature near the Pt heating circuit was very close to the heating temperature, and there existed a distinct temperature gradient along the direction toward the sensor tip, shown as the dark blue position in Figure 3. The temperature gradient was found to increase gradually with the increase in the heating temperature, and the lowest temperatures for each heating temperature were 178.6, 350.4, 529.5, and 705 °C at the end of the sensor. Additionally, the highest temperature points were located at the Pt heating circuit in the testing, which was attributed to the relatively high thermal conductivity. It was obvious that the temperature of the Pt heating circuit increased more rapidly than that of the ZrO$_2$ layer during the heating process. These results are coincident to the results reported in the literature by David Teyen Huang [24].

Similarly, the temperature gradients between layers were observed to be relatively smaller in the testing sensor. Figure 4 shows a 2D plot of the temperature gradients at sensor cross-sections, and uniform temperature distributions were observed except at several joint regions. This result indicated that the heat was evenly conducted inside of the sensor, which was probably determined by the structure and location of the Pt heating circuit. Considering that the Pt heating circuit was thin, with a thickness of 15 μm, it had a relatively wide surface of 3 × 40 mm, and the heat could rapidly transfer from metallic Pt

to adjacent ceramic layers owing to the large conducting area. However, steep temperature gradients were found to be located at the junctions between the Pt electrode and the ceramic layers, shown as the light blue parts of Figure 4. This implies that the interfaces between the layers will delay heat transfer. The maximum temperature gradients at the sensor cross-sections were 13.229, 28.348, 43.467, and 58.586 °C/mm at the heating temperatures of 200, 400, 600, and 800 °C respectively.

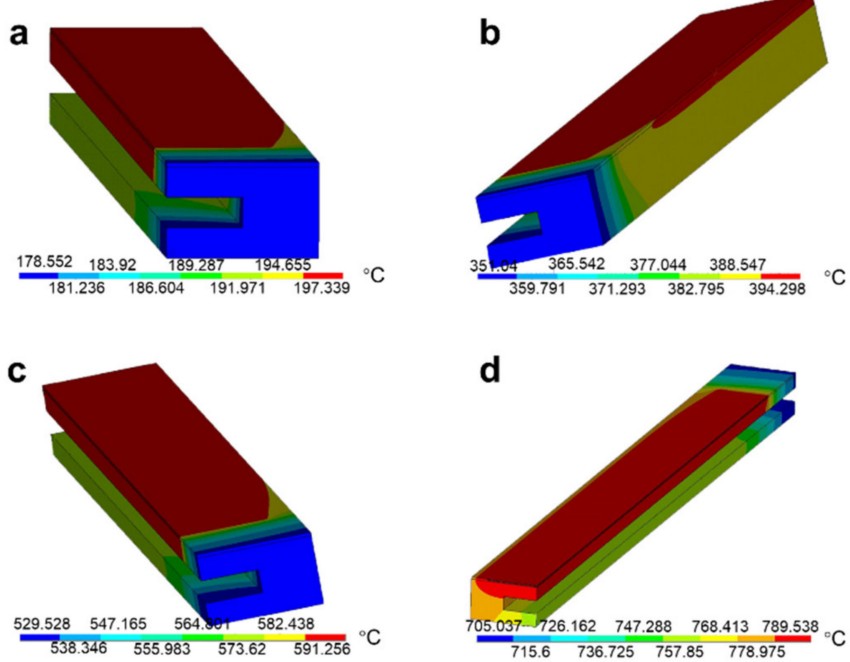

**Figure 3.** Temperature distribution of the nitrogen–oxygen sensors at different heating temperatures: (**a**) 200; (**b**) 400; (**c**) 600; (**d**) 800 °C.

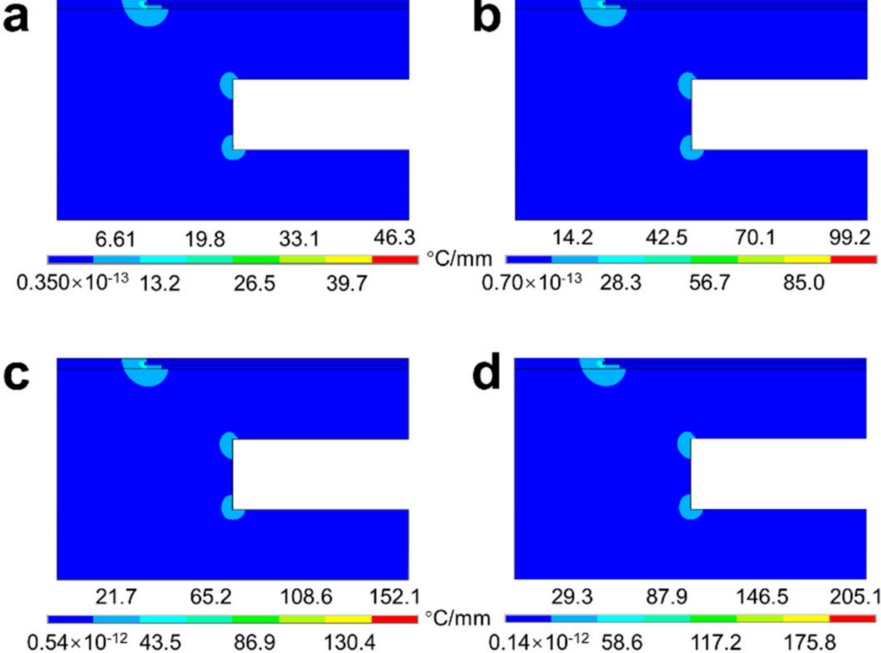

**Figure 4.** Temperature gradients at cross–sections of the nitrogen–oxygen sensors at different heating temperatures: (**a**) 200; (**b**) 400; (**c**) 600; (**d**) 800 °C.

Derived from the simulating temperature distribution results, Figure 5a–d represent the thermal deformation of the Pt heating circuit at 200, 400, 600, and 800 °C, which demonstrate the thermal deformation of the NO$_x$ sensor at the selected heating temperatures. It was observed that the thermal deformation presented a gradient distribution along the length direction of the sensor, represented by the direction of the arrow. The maximum thermal deformation position was coincident with the lowest temperature region. The maximum thermal deformations of the sensors were determined as 0.0389, 0.084, 0.129, and 0.174 mm corresponding to the heating temperatures 200, 400, 600, and 800 °C, respectively. Based on these results, it was found that maximum thermal deformation increased with the increase in heating temperature. Therefore, it could be concluded that the thermal deformation was correlated with the temperature gradient in the sensor.

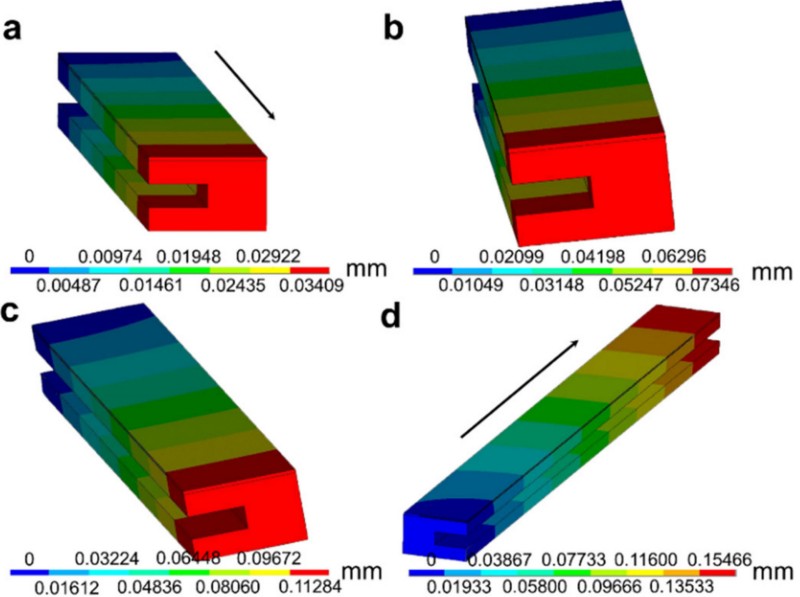

**Figure 5.** Thermal deformation distribution of the nitrogen–oxygen sensors under different heating temperatures: (**a**) 200; (**b**) 400; (**c**) 600; (**d**) 800 °C.

Figure 6 shows the thermal stress distributions of an NO$_x$ sensor under heating temperatures ranging from 200 to 800 °C. When the temperature was between 200 and 600 °C, the maximum thermal stresses were located at the connection between the left and right ends of the Pt heating circuit and the Al$_2$O$_3$ layer; however, when the temperature increased to 800 °C, the maximum thermal stress shifted to the interface between the Al$_2$O$_3$ layer and the ZrO$_2$ layer. At the selected heating temperatures, the maximum thermal stresses were observed as 141, 274, 407, and 540 MPa, respectively. Due to the higher thermal conductivity and thermal expansion coefficient of the Pt heating circuit, it was concluded that the high stress gradient was generated between the Pt heating circuit and the Al$_2$O$_3$ layer. During the heating process, the Pt heating circuit exhibited faster heating rates, resulting in obvious thermal volume expansion compared with the adjacent Al$_2$O$_3$ insulation layer. Moreover, the poor surface wettability between the metallic Pt and the ceramic Al$_2$O$_3$ layers also led to the weak interfacial bonding strength, increasing the probability of delamination between Pt and alumina.

The thermal stress concentration could be observed on the right-angle position of air chamber at above four simulated temperatures. The thermal stress was determined as 82.3, 177, 271, and 365 MPa with the corresponding temperatures of 200, 400, 600, and 800 °C, respectively. By comparing the temperature distributions, it was noted that the thermal stress was concentrated at the right-angle air chamber, which is usually the low-temperature and high-temperature gradient region in the sensor. As the thermal stress is proportional to the heating temperature, it was inferred that the thermal stress concentration was related

to the temperature gradient in the sensor. This can be understood in that the components in the sensor are subject to temperature gradients, leading to the generation of thermal deformation in the regions with high thermal stress concentrations.

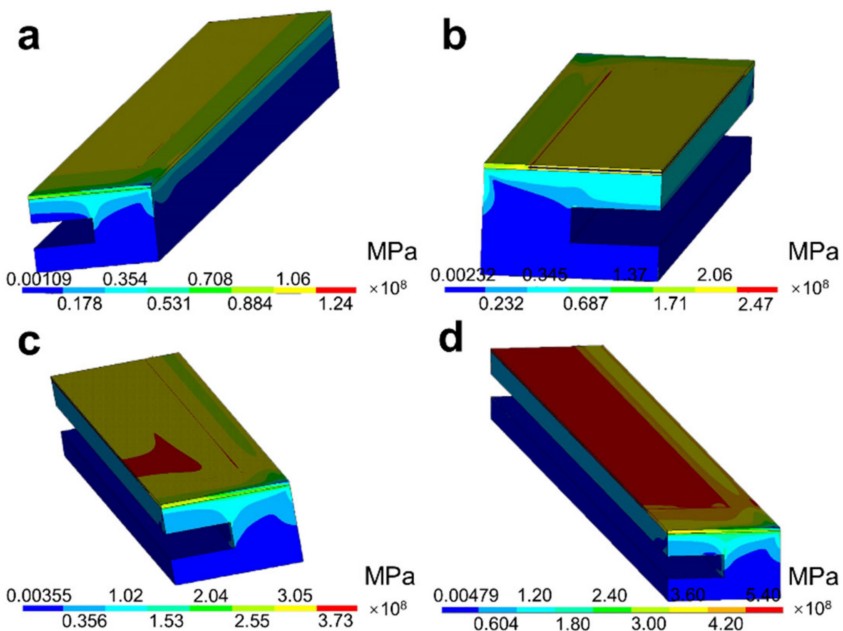

**Figure 6.** Thermal stress distribution of the nitrogen–oxygen sensors under different heating temperatures: (**a**) 200; (**b**) 400; (**c**) 600; (**d**) 800 °C.

### 3.2. Thermal Stress Failure Analysis of $NO_x$ Sensor

To verify the numerical simulation results of temperature and thermal stress distribution of the $NO_x$ sensor, the micromorphology of the failure sample resulting from thermal stress was examined by SEM and X-ray micro CT, observing the typical failure types and locations in the tested sensor. Figure 7a,b show cross-sectional micrographs of the $NO_x$ sensors obtained by SEM and X-ray micro CT. It can be observed that the defects mainly presented in the forms of the cracks and delamination, which were found surrounding the air chamber. Furthermore, the cracks were observed at the right-angle position of air chamber, as shown in Figure 7c,d, whereas the interfacial delamination occurred at the interface between the $Al_2O_3$ layer and the $ZrO_2$ layer, as shown in Figure 7e. These experimental results were coincident with the numerical simulation's predictions, indicating the higher stress concentrations generated on the abovementioned regions. According to the temperature and thermal stress simulation, as well as the microstructure examination of the $NO_x$ sensor samples, it can be concluded that structural defects are more likely to occur on the right angle of the air chamber during thermal cycle operations, which could further cause structural fractures of the sensor.

It can be confirmed that thermal gradients and thermal deformation contribute to the thermal stresses, inducing the formation of defects such as cracks and interface delamination inside of the sensor. This is the main reason for the decrease in tensile stress when the sensor was operating during thermal cycling. The cracks can easily extend to the surface and lead to fracture of the sensor, because the sensor is often vibrated in a working engine. The interface delamination will block heat transfer and form a local overheating region, resulting in the generation of thermal stress concentration. Based on the above results, some manufacturing processes can be improved to reduce the structural defects and achieve higher strength reliability during thermal cycling operation such as optimizing the sinter process of multilayer ceramics to enhance the interface bonding strength between the $Al_2O_3$ layer and the $ZrO_2$ layer/Pt heating circuit layer and applying reasonable structural treatment on the right angle of the air chamber to reduce thermal stress concentration.

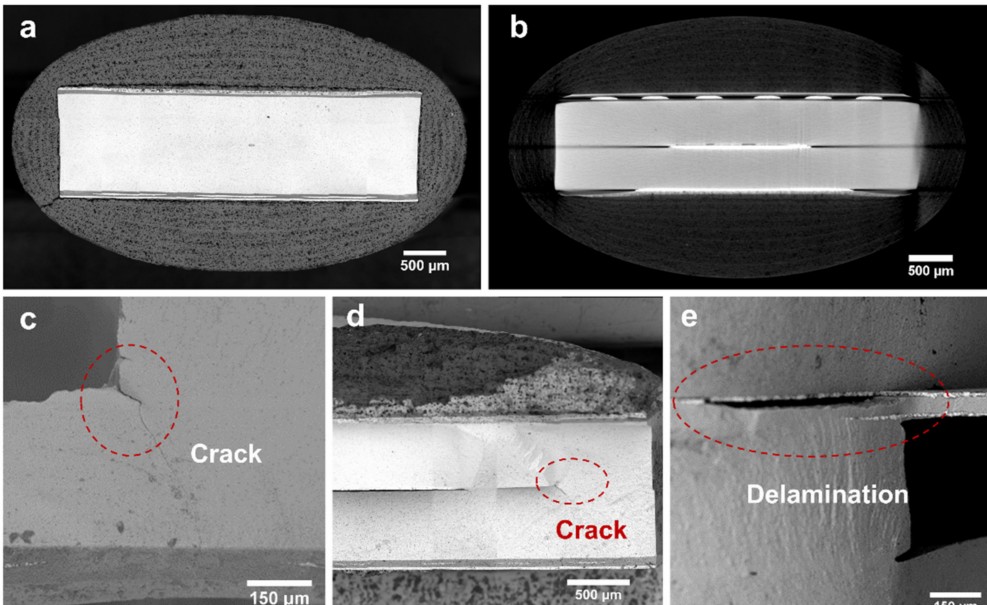

**Figure 7.** Micromorphology of nitrogen–oxygen sensors including SEM images (**a**) and X-ray micro CT image (**b**) of a cross-sectional view; cracks at the right-angle position of the air chamber (**c**,**d**); delamination position between the $Al_2O_3$ layer and the $ZrO_2$ layer (**e**).

## 4. Conclusions

In this project, we investigated the thermal stress and deformation of $NO_x$ sensors at different heating temperatures through numerical simulations. Furthermore, the failure model caused by thermal stress was confirmed by observing the location and type of the defect of the sample. The following conclusions were drawn:

(1) In the testing of $NO_x$ sensors, the temperature distribution near the Pt heating circuit was relatively uniform, and there was an obvious temperature gradient along the length direction that increased with the increase in heating temperature. In contrast, the temperature gradient along the interlayer was smaller owing to the higher heat transfer efficiency of the Pt heating circuit;

(2) The thermal deformation of the $NO_x$ sensor presented a gradient distribution along the length direction. The maximum thermal deformation location observed was consistent to that of the highest temperature gradient, which indicates that the temperature gradient had a significantly impact on the thermal deformation of the sensor;

(3) The overall thermal stress increased when the temperature increased. The maximum thermal stress was observed at the region of the Pt heating circuit and two $Al_2O_3$ layers below 600 °C, and the region changed to the interfaces between the $Al_2O_3$ layer and the $ZrO_2$ layer when the temperature reached 800 °C;

(4) Based on the simulations and microstructural observations, most of the defects were found around the air chamber in the form of cracks and delamination, which may cause decay in the mechanical strength and localized overheating of the sensors. It can be further confirmed that the cracks mainly occurred at the right-angle position of the air chamber, while the delamination occurred between the $Al_2O_3$ and the $ZrO_2$ layers.

**Author Contributions:** The following lists all authors' names and their contributions to the manuscript: Conceptualization, J.F. and J.G.; methodology, J.F. and J.G.; software, H.S.; validation, H.S.; formal analysis, J.F. and J.G.; investigation, J.F. and J.G.; resources, T.Z., B.C. and J.P.; data curation, J.F. and J.G.; writing—original draft preparation, J.F., J.G. and H.S.; writing—review and editing, T.Z., B.C. and J.P.; visualization, J.F. and J.G.; supervision, T.Z., B.C. and J.P.; project administration, J.P.; funding acquisition, B.C. and J.P. All authors have read and agreed to the published version of the manuscript.

**Funding:** This research was funded by the National Science Foundation of China (grant number: 52072135).

**Data Availability Statement:** Not applicable.

**Acknowledgments:** The authors would like to thank the National Science Foundation of China for the financial support, as the works were performed under contract (52072135). Thanks should also go to the Analytical and Testing Center of Huazhong University of Science where the SEM characterizations were completed.

**Conflicts of Interest:** The authors declare that they have no conflict of interest to this work. We declare that we do not have any commercial or associative interest that represents a conflict of interest in connection with the work submitted.

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
