# Peer review of "Thermal Stress Simulation and Structure Failure Analyses of Nitrogen–Oxygen Sensors under a Gradual Temperature Field"

_energies, doi:10.3390/en15082799_

Round 1
Reviewer 1 Report
This is an interesting paper reporting results of finite element modelling of thermally-induced stresses and potential fracture formation in Nitrogen-Oxygen sensors under Gradual Temperature Field. The paper is rather well presented. The results are certainly significant both from a scientific viewpoint and in view of improvements in the nitrogen-oxygen sensors design and manufacturing. I therefore tend to recommend this paper for publication on Energies. However, I think that the manuscript needs to be improved in a few significant points, that I summarise in the list here below:
- Introduction: I would recommend to add a paragraph in which the structure and functioning of solid-electrolyte nitrogen-oxygen sensors is better clarified for the reader. In the current form, the average reader will only understand the use of yttria-stabilised zircona as a solid electrolyte for the quantification of oxygen content, but not how the latter relates to the concentration of NO or NO2 in the exhaust gas. I think the presentation of the paper would be largely improved by better detailing how the sensors actually measure both NO and NO2 concentrations, as this will allow the reader to have a more comprehensive understanding also of the implications of thermal stresses and formation of cracks on the global performance of these sensors. This point might be elegantly recalled then in the conclusions section (see below).
- Input parameters to the calculations are presented in Table 1 and Figure 2, and results are summarised in Figures 3, 4 and 5. I also appreciate the comparison with electron microscopy images reported in Figure 6, which give the computational results some quite convincing soundness. However, it is impossible for me - and for any reader- to assess the quality and accuracy of the present calculations without checking the essential equations of the model. I am sure the model is based on heat flow, thermal expansion and stress-strain relationship in the materials. However, these equations can become rather complicated when it comes to composite materials and boundary conditions and the interfaces. I recommend the Authors to please add a section reporting the essential equations and boundary conditions of the model, where critical parameters affecting the calculation accuracy are clearly identified.
- In Section 3.2, some technical details should be added about the techniques with which images presented in Figure 6 were obtained: type of microscope, potential precision, etc.
- In the Conclusions section, I wonder if the Authors could add a point in which they estimate which implications the current results on nitrogen-oxygen sensor stresses and cracks may have on the performance of the sensor itself. This point may relate to the more general functioning of a N-O sensor, which I recommend to add to the introduction section and mentioned in point 1. above.
Reviewer 2 Report
The authors introduced the structure parameters of planar NO x sensor and using the thermal stress finite element analysis program, the relationship between the thermal stress distribution and operation temperature can be revealed, which can provide theoretical support for the structural designs and functional optimizations for the future NOx sensors. Based on the numerical simulation results, the type and location of the defect forming inside of multilayers ceramics were predicted.
The manuscript could be accepted after major revision.
The literature should be supported by published articles in energies.
The abstract and introduction should be improved.
What is the used FE software?
Why does the temperature distribution near the Pt heating circuit was relatively uniform?
The overall thermal stress increases when the temperature increases, HOW?
The quality of (Fig 1) is very poor.
The used units should be added in Fig 3, 4 and 5.
The discussions are very weak.
Why does the delamination occurred at the interface between Al2O3 layer and ZrO2 layer.
The thermal properties of the used materials should be included.
Round 2
Reviewer 1 Report
The Authors have now addressed all the comments and suggestions of the previous review. The paper can be published in its current form in my opinion, except for some moderate English corrections.
Author Response
Thank you very much! We have polished the language as best as we can.
Reviewer 2 Report
This is an interesting well-written paper reporting results of finite element modeling of thermal stresses in Nitrogen-Oxygen sensors. The paper could be accepted after a minor revision.
- A 2D plots of temperature gradients must be included.
- You used 200 °C, 400 °C, 600 °C and 800 °C, why? And could you provide a regression model o predict the responses at different temperatures?
- Strength your introduction by thermal deflection and thermal stresses in a thin circular plate under an axisymmetric heat source or modeling of cooling and heat conduction in permanent mold casting process and others.
- T = T (x, y, z), it should be modified.
- Mesh size and type should be included.
